∂ | **Open Peer Review** | Bacteriology | Research Article

# Rapid identification and methicillin resistance test to *Staphylococcus aureus* in cerebrospinal fluid by MALDI-TOF MS

Mengyu Zhang,[1] Xuanxuan Wang,[2] Wei Huang,[3] Ailing Ma,[1] Shuguo Qin,[1] Di Hu,[1] Henggui Hu,[1] Xiaolei Du,[1] Kaixuan Zhang,[1] Sudi Zhu,[1] Yuanyuan Xu[1]

**ABSTRACT**    Bacterial meningitis is a disease with high mortality and morbidity, and it primarily manifests as symptoms involving the central nervous system (CNS). Hence, it would be of great importance to make an early diagnosis and initiate empirical antimicrobial treatment in time for this disease. In this study, we investigated the feasibility of rapid *Staphylococcus aureus* (*S. aureus*) identification and drug resistance analysis through the combination of centrifugation-based enrichment of bacteria and matrix-assisted laser desorption ionization time-of-flight mass spectrometry (MALDI-TOF MS). Specifically, the cerebrospinal samples were treated by differential centrifugation to collect precipitates after a short-term rapid culture. Then, the precipitates were identified by MALDI-TOF MS. Subsequently, the bacterial solution ($10^6$ CFU/mL) was mixed with an equal volume of cation-adjusted Mueller-Hinton broth (CAMHB) supplemented with cefoxitin (4 µg/mL). After the culture of the mixture, the precipitates obtained by centrifugation were analyzed by MALDI-TOF MS. The efficiency of centrifugation-based enrichment of *S. aureus* was 87.9% at $10^3$ CFU/mL and increased to 90.4% at $10^2$ CFU/mL. This identification efficiency reached 100% after an 8-h culture. The optimal testing time for bacterial resistance identification was achieved by the culture within 3 h. The validity, sensitivity, and specificity were all 100% at this time point. The results of the rapid identification method were identical to those of the broth microdilution method. Through this protocol, the identification and drug resistance analysis of *S. aureus* in the cerebrospinal fluid (CSF) can be completed within 11 h. These findings are expected to provide a new method for the rapid diagnosis and treatment of patients with bacterial meningitis.

**IMPORTANCE**  This study introduces a novel method combining centrifugation-based bacterial enrichment with MALDI-TOF MS, enabling an 11-h identification of *Staphylococcus aureus* and direct detection of methicillin resistance directly from the CSF. Validated with 100% sensitivity and specificity, this approach accelerates targeted antibiotic therapy and establishes a streamlined workflow for rapid diagnosis and treatment of bacterial meningitis.

**KEYWORDS**  cerebrospinal fluid, MALDI-TOF MS, antibiotic resistance, *S. aureus* identification, MRSA

Bacterial meningitis is a worldwide health problem characterized by rapid onset, high mortality and morbidity, and high outbreak and epidemic potentials (1–3). This disease is caused by bacterial invasion of the meninges and primarily manifests as symptoms involving the central nervous system (CNS), with *Staphylococcus aureus* (*S. aureus*) being one of the causative organisms (4). *S. aureus* colonizes about 20 to 80% of the human population and provides a reservoir for subsequent transmission and infection (5). Globally, *S. aureus* causes 1–9% of bacterial meningitis cases with antibiotic

Address correspondence to Kaixuan Zhang, 2461530536@qq.com, Sudi Zhu, 2638417678@qq.com, or Yuanyuan Xu, 2097788385@qq.com.

Mengyu Zhang and Xuanxuan Wang contributed equally to this article. Author order was determined alphabetically.

The authors declare no conflict of interest.

See the funding table on p. 11.

10.1128/spectrum.01508-25  **1**

resistance (notably MRSA) and high mortality rates (14–56%) (4). At our institution (Wanbei Coal Electric Group General Hospital, Anhui, China), empirical data indicate *S. aureus* as a predominant gram-positive pathogen in meningitis diagnostics, though precise institutional incidence requires expanded surveillance. The proliferation of multidrug-resistant strains imposes constraints on the selection of antibiotic treatment options, thereby posing challenges to clinical intervention (3, 6, 7). Identifying bacterial meningitis pathogens and analyzing drug resistance constitute the key to achieving effective clinical treatment.

Meningitis cannot be diagnosed based on clinical features alone, and the cerebrospinal fluid (CSF) analysis is essential for its diagnosis. The microscopic examination and microbial culture of CSF samples constitute the gold standard for the diagnosis of this disease. It is essential to make an early diagnosis and initiate empirical antimicrobial treatment in time for this disease (8–10). However, conventional bacterial cultures and drug susceptibility tests typically require 3–5 days, leading to delays in treatment. Existing techniques, such as molecular diagnostic technology, can be employed to rapidly identify pathogens and detect drug resistance. However, such potential factors as inhibitors and cross-contamination may result in erroneous microorganism identification (11). Therefore, there is an urgent demand for simpler and more cost-effective detection methods.

Currently, matrix-assisted laser desorption ionization time-of-flight mass spectrometry (MALDI-TOF MS), which features simple procedures, fast detection, and low costs, is a common method for the clinical detection of bacteria (12–14). Recent studies have explored the potential of MALDI-TOF MS for detecting bacterial drug resistance, though its clinical application remains investigational. Rhoads et al. reported that the MALDI-TOF mass spectral peak could be used to predict methicillin resistance in staphylococci (15). Nix ID et al. developed a direct targeted microdrop growth assay (DOT-MGA) based on MALDI-TOF MS to rapidly detect bacterial resistance within 5 h (16). This method allows for the detection of methicillin-resistant *S. aureus* from both blood culture flasks and solid culture media (17). In urinary tract infections, the DOT-MGA method can also be utilized to detect drug-resistant bacteria (18). However, the detection of drug-resistant bacteria in the CSF has not been reported.

In this study, mass spectrometry based on centrifugal enrichment was used to rapidly identify *S. aureus* and detect drug resistance, in an attempt to minimize turnaround time and provide a foundation for precise clinical treatment.

## MATERIALS AND METHODS

### Bacterial strains

A total of 40 *S. aureus* isolates were included in this study, including 20 MRSA and 20 MSSA strains. All isolates were collected from the routine diagnostics at Wanbei Coal-Electricity Group General Hospital (Anhui, China), including CSF from patients with bacterial meningitis and other clinical sites (sputum, mucosal secretions, and cutaneous secretions) (Table S10). To avoid isolating duplicates, only 1 isolate per patient was included. The *mecA* gene was detected in all MRSA isolates (Table S10). The mecA was detected via PCR using species-specific primers (mecA-F/R), with reaction conditions including 35 cycles of denaturation (95°C, 30s), annealing (58°C, 30s), and extension (72°C, 30s). Amplicons were verified by 1% agarose gel electrophoresis and Sanger sequencing (Sangon Biotech, China), with sequences aligned to NCBI GenBank for confirmation. The sequencing data of the mecA gene from *S. aureus* generated in this study are available at the following link: https://doi.org/10.5281/zenodo.18062160. The minimum inhibitory concentration (MIC) of 40 samples was obtained by the micro broth dilution method.

## Bacterial stock preparation

The bacteria were inoculated on a blood culture plate at 35°C for 12 hours. To simulate clinical infection samples, single colonies were suspended in CSF and vortexed. The CSF was obtained from patients with confirmed bacterial culture-negative CSF samples and solely collected via lumbar puncture. Subsequently, the absorbance of the bacterial solution at 600 nm was measured by ultraviolet-visible spectroscopy (UV-VIS) and adjusted to 0.7 to 0.8 as the bacterial stock solution. The bacterial concentration was determined by the plate counting method.

## MIC determination

The MIC of cefoxitin was determined for *S. aureus* using the broth microdilution reference method according to the Clinical and Laboratory Standards Institute (CLSI) (19) and International Organization for Standardization (ISO) guidelines (20). All antibiotic dilutions and panel preparations were performed in-house. The bacterial colonies on the blood agar plate were transferred into sterile water and mixed. Then, the $OD_{600}$ of the bacterial suspension was measured using UV-VIS and adjusted to a range of 0.7 to 0.8. Subsequently, the bacterial suspension was diluted with the cation-adjusted Mueller-Hinton broth (CAMHB) to achieve a final concentration of $10^6$ CFU/mL. To validate the accuracy of $OD_{600}$-based turbidity adjustment, the serial dilution was inoculated onto triplicate nutrition agar plates (Tianda Reagent, China), and the colonies were enumerated after an overnight culture. The bacterial suspension (100 µL) was diluted with cefoxitin at a ratio of 1:2 in a 96-well plate to achieve a final concentration of $5 \times 10^5$ CFU/mL, and the final concentration of cefoxitin was increased in a doubling manner, ranging from 0.5 to 256 µg/mL. The results were read after the culture for $18 \pm 2$ h at 35°C. The MIC50, MIC90, and MIC ranges were calculated for further analysis.

## Centrifugation-based enrichment efficiency

The bacterial stock solution was diluted 10 times with sterile water, resulting in a concentration of $10_6$ CFU/mL. Then, the bacterial suspension was diluted 10 times with sterile water until reaching a final concentration of $10^2$ CFU/mL. The diluted bacterial suspension was centrifuged at 12,000 rpm for 5 min, and the supernatant and precipitate were streaked onto nutrient agar plates at 35°C for 12 h. The accurate bacterial concentration was determined by the plate counting method, and the enrichment efficiency (%) was calculated as: (CFU in precipitate) / (CFU in precipitate + CFU in supernatant) × 100. The bacterial suspension was diluted to a concentration of 10 CFU/mL, and the above steps were repeated in triplicate to observe the efficiency of centrifugation-based enrichment.

## Rapid identification of *S. aureus* in the CSF

The bacterial stock solution was diluted to $10^4$ CFU/mL with sterile water, and the bacterial suspension (1 µL) was added to the Luria-Bertani (LB) broth (9, 19, and 39 µL) in varying volumes. The CSF was utilized as a growth control. Then, the mixture was incubated at 35°C under shaking at 900 rpm for 8 h, and the $OD_{600}$ was measured every hour. When the $OD_{600}$ was >0.8, the culture was stopped, and the mixture was adjusted to 0.7–0.8 with sterile water. When the $OD_{600}$ was 0.7–0.8, the mixture was centrifuged at 12,000 rpm for 5 min, followed by supernatant removal and washing with 100 µL sterile water three times as well as resuspension in 5 µL of 70% formic acid and 5 µL of acetonitrile (ACN). Subsequently, the analysis was performed based on MALDI-TOF MS. The optimal volume of the LB broth was determined by comparing the $OD_{600}$ values of three groups with varying volumes. Finally, 40 strains of *S. aureus* were rapidly identified.

## Bacterial resistance analysis of the CSF by MALDI-TOF MS

The bacterial stock solution was diluted to a concentration of $5 \times 10^5$ CFU/mL with CAMHB. Subsequently, different volumes of the bacterial suspension (10, 20, 40, 80, and

160 µL) were incubated at 35°C under shaking at 900 rpm, and the $OD_{600}$ was measured every hour (1 to 4 h). After the measurement, the bacterial suspension was centrifuged at 12,000 rpm for 5 min, followed by supernatant removal and washing with sterile water three times, as well as resuspension in 5 µL of 70% formic acid and 5 µL of ACN. Then, the analysis was performed based on MALDI-TOF MS. The optimal volume of CAMHB was determined by comparing the $OD_{600}$ values of five groups with varying volumes.

After a short-term culture for bacterial identification, 10 µL of bacterial suspension was mixed with 990 µL of CAMHB solution. Based on the above experimental results, the optimal volume for the bacterial culture was determined to be 40 µL. Cefoxitin was dissolved in the CAMHB solution at a concentration of 8 µg/mL, and then 40 µL of Cefoxitin solution was added to the diluted bacterial suspension (40 µL) in CAMHB. Eventually, an expected final bacterial inoculation volume of approximately 5 × $10^5$ CFU/mL was obtained. The final concentration of cefoxitin reached 4 µg/mL. For this experiment, one MRSA strain and one MSSA strain were selected. Cefoxitin was added to the bacterial suspension as the experimental group, while CAMHB without the addition of antibiotics served as the control group. The bacteria were incubated at 35°C under shaking at 900 rpm for 8 h, with $OD_{600}$ values measured every hour in this period. At the end of the culture, the sample was centrifuged at 12,000 rpm for 5 min, followed by supernatant removal and washing with sterile water three times. The analysis was performed based on MALDI-TOF MS. The remaining strains were measured by the above method.

## MALDI-TOF MS identification

The bacterial protein solution (1 µL) treated with 70% formic acid and ACN was applied onto the MALDI-TOF MS target plate, followed by the addition of theα-cyano-4-hydroxy-cinnamic acid (CHCA) matrix (1 µL, 10 mg/mL in ACN/H2O [vol/vol = 1/1] containing 2% trifluoroacetic acid [TFA]) and subsequent drying for MALDI-TOF MS analysis. MALDI-TOF MS analysis was performed using a MALDI Biotyper System (Bruker Daltonics, Germany) operated in linear positive ion mode with a mass range of 2,000–20,000 Da. Spectral acquisition and processing were conducted using Bruker Biotyper software with the MBT RUO-DB8468 database. Raw spectra were processed with standard parameters, including spectral smoothing, baseline correction, and automatic peak detection. If the species identification score for the growth control without cefoxitin was ≥1.7, the test was considered valid; a score of < 1.7 indicated an invalid test. For samples with cefoxitin, successful species identification (score ≥ 1.7) was interpreted as a non-susceptible result for the given isolate, whereas failed species identification (score < 1.7) was classified as a susceptible isolate. A median result for three spots was calculated and used for further analysis.

## Statistical analysis

SPSS 26.0 was utilized to conduct a statistical credit analysis. The continuous calibration $\chi^2$ test ratio was employed to demonstrate the effectiveness rate. The results were expressed as mean ± standard deviation (SD) or percentage. The consistency of the DOT-MGA and microbroth dilution method was evaluated using the Kappa test. A Kappa value of 1 indicated identical results; a Kappa value between 1 and 0.75 indicated good consistency; a Kappa value between 0.75 and 0.4 indicated general consistency; and a Kappa value below 0.4 indicated poor consistency. The significance level ($a$) for testing was set at 0.05.

## RESULTS

### Enrichment efficiency of the bacterial solution

The enrichment efficiency of the bacterial solution was determined by centrifugation. As illustrated in Fig. 1A through D, in bacterial solutions of varying concentrations,

the enrichment efficiency was measured at 87.9 ± 1.4 and 90.4 ± 1.8%, indicating the successful extraction of the bacteria solution. The remaining data are shown in Fig. S1.

## Rapid identification of *S. aureus* through a microbroth culture in the CSF

The identification of bacteria was achieved through a microbroth culture, as illustrated in Fig. 2. In the CSF without the LB broth, there was no significant change in the $OD_{600}$ value after an 8-h culture. The $OD_{600}$ value was 1.12 ± 0.10, 0.89 ± 0.18, and 0.66 ± 0.14 when the samples were cultured with 9, 19, and 39 µL of the LB broth for 7 h, respectively. The changes in the $OD_{600}$ value are shown in Table S1. The LB broth of 9 µL exhibited the highest absorbance ever recorded. The specific $OD_{600}$ values of 40 strains of *S. aureus* cultured in 9 µL of LB broth are depicted in Table 1. When the $OD_{600}$ value was >0.8, the culture could be stopped for bacterial identification.

## Characterization of bacterial isolates using standard methods

As shown in Table S2, for MRSA isolates, the MIC50, MIC90, and MIC of cefoxitin ranged from 16, 128, and 8 to >256 mg/mL, respectively. For MSSA isolates, the MIC50, MIC90, and MIC of cefoxitin ranged from 2, 4, and 2 to 4 µg/mL, respectively. The MIC of the QC strain *S. aureus* (ATCC 25923) was within the recommended range throughout the study. The expected QC organism range is 1–4 µg/mL, which is based on the standards outlined in CLSI M100 (Performance Standards for Antimicrobial Susceptibility Testing). The *mecA* gene was detected in all clinical MRSA isolates.

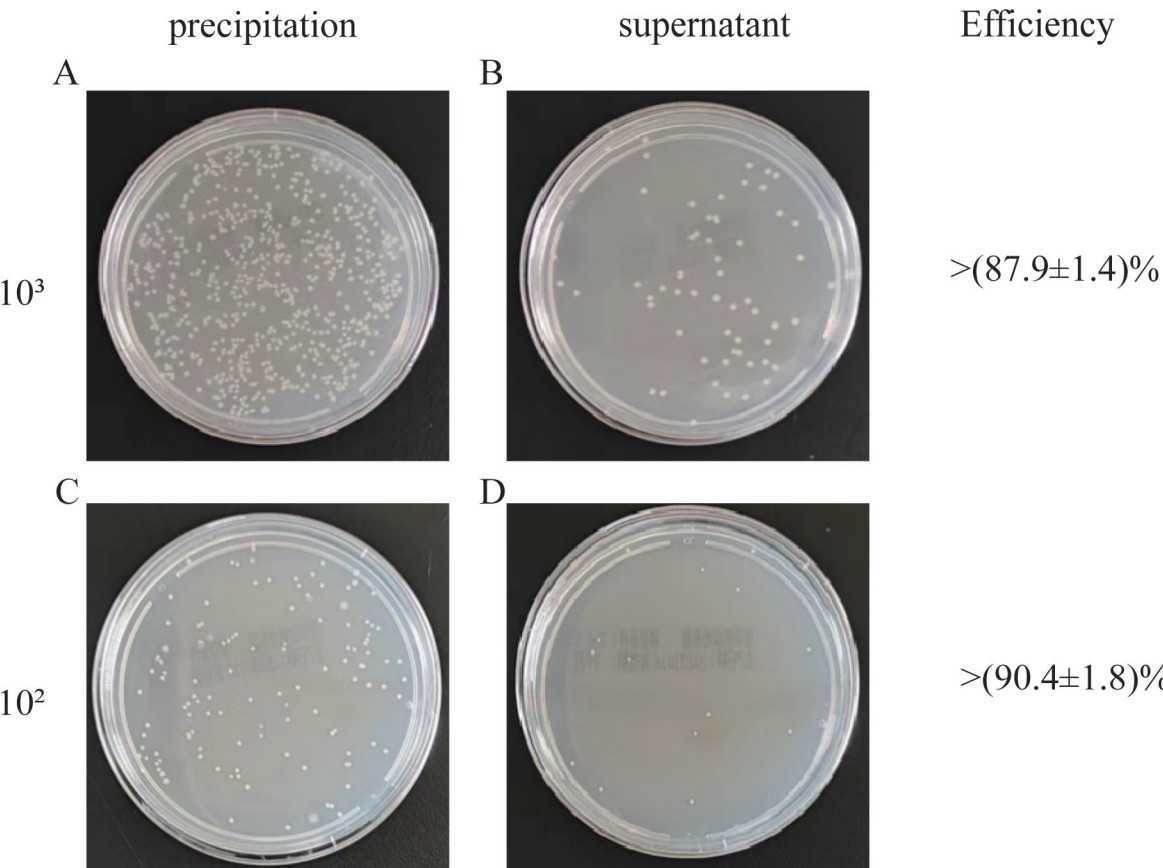

**FIG 1** Enrichment efficiency of centrifugation. Photographs of culture plates of the enriched pellets and supernatant for *S. aureus* within bacterial solutions of varying concentrations: $10^3$ (A and B) and $10^2$ CFU/mL (C and D).

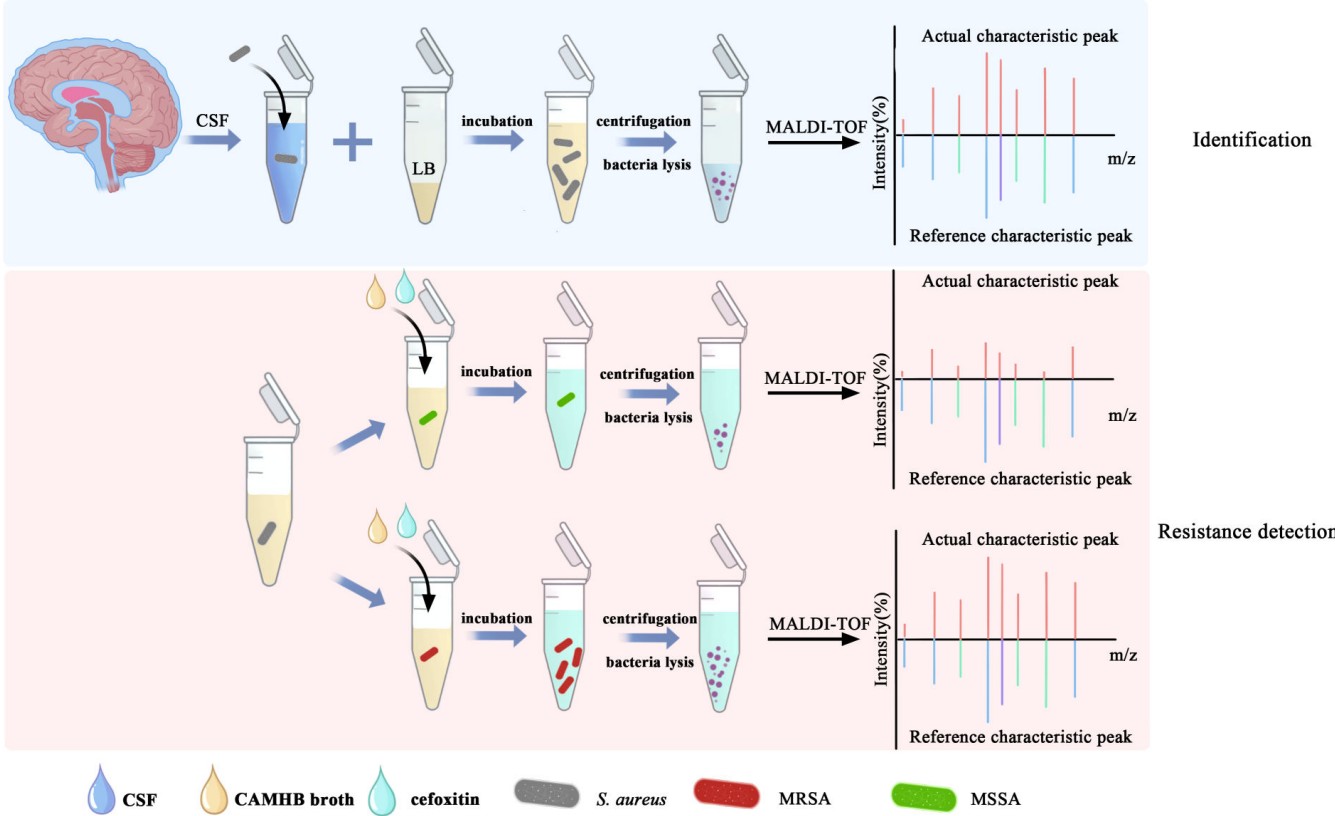

**FIG 2** Workflow for identification and drug resistance analysis of *S. aureus* in cerebrospinal samples.

## Rapid antimicrobial susceptibility testing by MALDI-TOF MS

The percentage of bacterial identification achieved through the culture based on various volumes is presented in Table S3. When cultured for 2 h, the identification scores for bacteria cultured using CAMHB with a volume of 40 μL were nearly indistinguishable from those obtained with a volume of 80 and 160 μL but superior to those obtained with a volume of 10 and 20 μL. This trend persisted after a 3-h culture. Without affecting the results of the experiment, the bacterial solutions with smaller volumes were selected in this study. Thus, 40 μL was selected as the optimal culture volume for the CAMHB bacterial suspension.

As shown in Fig. 3, MRSA exhibited a slow growth rate in the CAMHB broth with cefoxitin (4 μg/mL), and a significant difference in $OD_{600}$ was observed between the control and drug groups after a 7-h culture ($P < 0.036$). Conversely, MSSA failed to proliferate in the CAMHB containing cefoxitin (4 μg/mL). There was a significant difference in $OD_{600}$ values between the control and drug groups after a 2-h culture ($P < 0.001$). The $OD_{600}$ values are presented in Table S4. Based on previous experiments (Table S3), most of the bacteria can be identified after a 3-h culture. Therefore, to achieve accurate identification, the detection time lengths of 2, 3, and 4 h were selected for the drug resistance experiment.

As shown in Table S5, MRSA grew faster in the CAMHB without cefoxitin, with the $OD_{600}$ values reaching 0.08 ± 0.03, 0.28 ± 0.07, and 0.76 ± 0.10 after a 2-, 3-, and 4-h culture, respectively. MRSA exhibited a slower growth rate in CAMHB (4 μg/mL cefoxitin), with the $OD_{600}$ values reaching 0.07 ± 0.03, 0.24 ± 0.09, and 0.41 ± 0.18, respectively, at the same time points. As shown in Table S6, the $OD_{600}$ of MSSA was 0.08 ± 0.03, 0.32 ± 0.07, and 0.72 ± 0.12 when cultured in CAMHB for 2, 3, and 4 h, respectively; the $OD_{600}$ of MSSA was 0.02 ± 0.01, 0.02 ± 0.01, and 0.02 ± 0.01 when cultured in CAMHB (4 μg/mL cefoxitin) for 2, 3, and 4 h, respectively.

**TABLE 1** $OD_{600}$ value changes during *S. aureus* identification[a]

| Strain | 4 h | 5 h | 6 h | 7 h | 8 h |
|---|---|---|---|---|---|
| R-1 | 0.04 ± 0.01 | 0.15 ± 0.04 | 0.46 ± 0.04 | 1.17 ± 0.08 | – |
| R-2 | 0.02 ± 0.01 | 0.06 ± 0.02 | 0.13 ± 0.02 | 0.91 ± 0.02 | – |
| R-3 | 0.02 ± 0.01 | 0.05 ± 0.02 | 0.09 ± 0.01 | 0.77 ± 0.05 | 1.16 ± 0.09 |
| R-4 | 0.03 ± 0.01 | 0.13 ± 0.04 | 0.38 ± 0.04 | 1.12 ± 0.10 | – |
| R-5 | 0.01 ± 0.00 | 0.02 ± 0.01 | 0.04 ± 0.02 | 0.32 ± 0.02 | 0.79 ± 0.04 |
| R-6 | 0.02 ± 0.01 | 0.02 ± 0.01 | 0.13 ± 0.01 | 0.86 ± 0.04 | – |
| R-7 | 0.03 ± 0.01 | 0.14 ± 0.03 | 0.27 ± 0.03 | 0.69 ± 0.04 | 1.88 ± 0.16 |
| R-8 | 0.02 ± 0.01 | 0.02 ± 0.01 | 0.17 ± 0.02 | 0.53 ± 0.03 | 1.42 ± 0.07 |
| R-9 | 0.01 ± 0.00 | 0.01 ± 0.00 | 0.08 ± 0.01 | 0.89 ± 0.05 | – |
| R-10 | 0.02 ± 0.01 | 0.05 ± 0.02 | 0.09 ± 0.01 | 0.82 ± 0.04 | – |
| R-11 | 0.01 ± 0.00 | 0.01 ± 0.01 | 0.07 ± 0.01 | 0.69 ± 0.03 | 1.14 ± 0.06 |
| R-12 | 0.03 ± 0.01 | 0.07 ± 0.02 | 0.13 ± 0.01 | 0.86 ± 0.04 | – |
| R-13 | 0.04 ± 0.02 | 0.11 ± 0.03 | 0.24 ± 0.03 | 1.10 ± 0.09 | – |
| R-14 | 0.01 ± 0.01 | 0.01 ± 0.00 | 0.04 ± 0.01 | 0.26 ± 0.02 | 0.91 ± 0.05 |
| R-15 | 0.01 ± 0.00 | 0.01 ± 0.00 | 0.03 ± 0.01 | 0.35 ± 0.02 | 1.28 ± 0.08 |
| R-16 | 0.02 ± 0.01 | 0.06 ± 0.02 | 0.12 ± 0.01 | 0.84 ± 0.04 | – |
| R-17 | 0.01 ± 0.00 | 0.04 ± 0.01 | 0.08 ± 0.01 | 0.58 ± 0.03 | 1.79 ± 0.20 |
| R-18 | 0.02 ± 0.01 | 0.02 ± 0.01 | 0.11 ± 0.01 | 0.29 ± 0.02 | 0.86 ± 0.04 |
| R-19 | 0.01 ± 0.00 | 0.01 ± 0.00 | 0.03 ± 0.01 | 0.14 ± 0.01 | 0.51 ± 0.03 |
| R-20 | 0.01 ± 0.01 | 0.02 ± 0.01 | 0.07 ± 0.01 | 0.33 ± 0.02 | 0.92 ± 0.04 |
| S-1 | 0.04 ± 0.01 | 0.17 ± 0.03 | 0.36 ± 0.05 | 1.75 ± 0.11 | – |
| S-2 | 0.02 ± 0.01 | 0.06 ± 0.02 | 0.11 ± 0.02 | 0.78 ± 0.05 | 1.51 ± 0.03 |
| S-3 | 0.01 ± 0.01 | 0.04 ± 0.01 | 0.09 ± 0.01 | 0.70 ± 0.04 | 1.18 ± 0.06 |
| S-4 | 0.02 ± 0.01 | 0.03 ± 0.01 | 0.14 ± 0.02 | 0.85 ± 0.05 | – |
| S-5 | 0.01 ± 0.01 | 0.06 ± 0.01 | 0.12 ± 0.02 | 0.69 ± 0.03 | 0.97 ± 0.05 |
| S-6 | 0.04 ± 0.01 | 0.19 ± 0.04 | 0.38 ± 0.05 | 1.36 ± 0.08 | – |
| S-7 | 0.02 ± 0.01 | 0.02 ± 0.01 | 0.06 ± 0.01 | 0.54 ± 0.03 | 1.83 ± 0.11 |
| S-8 | 0.03 ± 0.01 | 0.14 ± 0.03 | 0.37 ± 0.05 | 1.52 ± 0.08 | – |
| S-9 | 0.01 ± 0.01 | 0.04 ± 0.01 | 0.09 ± 0.01 | 0.38 ± 0.02 | 0.96 ± 0.04 |
| S-10 | 0.01 ±0.00 | 0.03 ± 0.01 | 0.08 ± 0.01 | 0.21 ± 0.02 | 1.02 ± 0.05 |
| S-11 | 0.04 ± 0.01 | 0.16 ± 0.03 | 0.42 ± 0.06 | 1.77 ± 0.09 | – |
| S-12 | 0.02 ± 0.01 | 0.03 ± 0.01 | 0.08 ± 0.01 | 0.36 ± 0.02 | 1.13 ± 0.05 |
| S-13 | 0.02 ± 0.01 | 0.04 ± 0.01 | 0.09 ± 0.02 | 0.58 ± 0.03 | 0.91 ± 0.04 |
| S-14 | 0.03 ± 0.01 | 0.11 ± 0.02 | 0.26 ± 0.03 | 1.17 ± 0.06 | – |
| S-15 | 0.02 ± 0.01 | 0.09 ± 0.02 | 0.18 ± 0.03 | 0.67 ± 0.04 | 1.64 ± 0.17 |
| S-16 | 0.01 ± 0.01 | 0.04 ± 0.01 | 0.08 ± 0.01 | 0.40 ± 0.02 | 0.88 ± 0.04 |
| S-17 | 0.02 ± 0.01 | 0.08 ± 0.02 | 0.16 ± 0.02 | 0.34 ± 0.02 | 1.27 ± 0.07 |
| S-18 | 0.04 ± 0.02 | 0.19 ± 0.03 | 0.44 ± 0.05 | 1.77 ± 0.09 | – |
| S-19 | 0.01 ± 0.01 | 0.05 ± 0.01 | 0.12 ± 0.02 | 0.42 ± 0.02 | 0.93 ± 0.04 |
| S-20 | 0.02 ± 0.01 | 0.03 ± 0.01 | 0.08 ± 0.01 | 0.44 ± 0.02 | 1.22 ± 0.11 |

[a]The dash (–) indicates that the OD value of the liquid medium was >0.8, and incubation stopped.

The identification of 40 strains of *S. aureus* in CAMHB (4 µg/mL cefoxitin) was performed using MALDI-TOF MS after the bacterial culture was completed, as presented in Table 2. Based on different enrichment methods, the bacteria were categorized into the non-centrifuge and centrifuge groups. For the non-centrifuge group, 20 strains of MRSA were not successfully identified after a 2-h culture. After a 3-h culture, 13 strains of MRSA were not identified successfully. Besides, three strains of MRSA were not successfully identified after a 4-h culture. In the centrifuge group, 12 strains of MRSA were not identified successfully after a 2-h culture. All strains were successfully identified after a 3- and 4-h culture. In contrast, the strains of MSSA were not identified successfully after a 2-, 3-, and 4-h culture in both groups (Table S7). The MALDI-TOF MS spectra of MSSA and MRSA are shown in Fig. 4. As presented in Table 3, the validity, sensitivity, specificity,

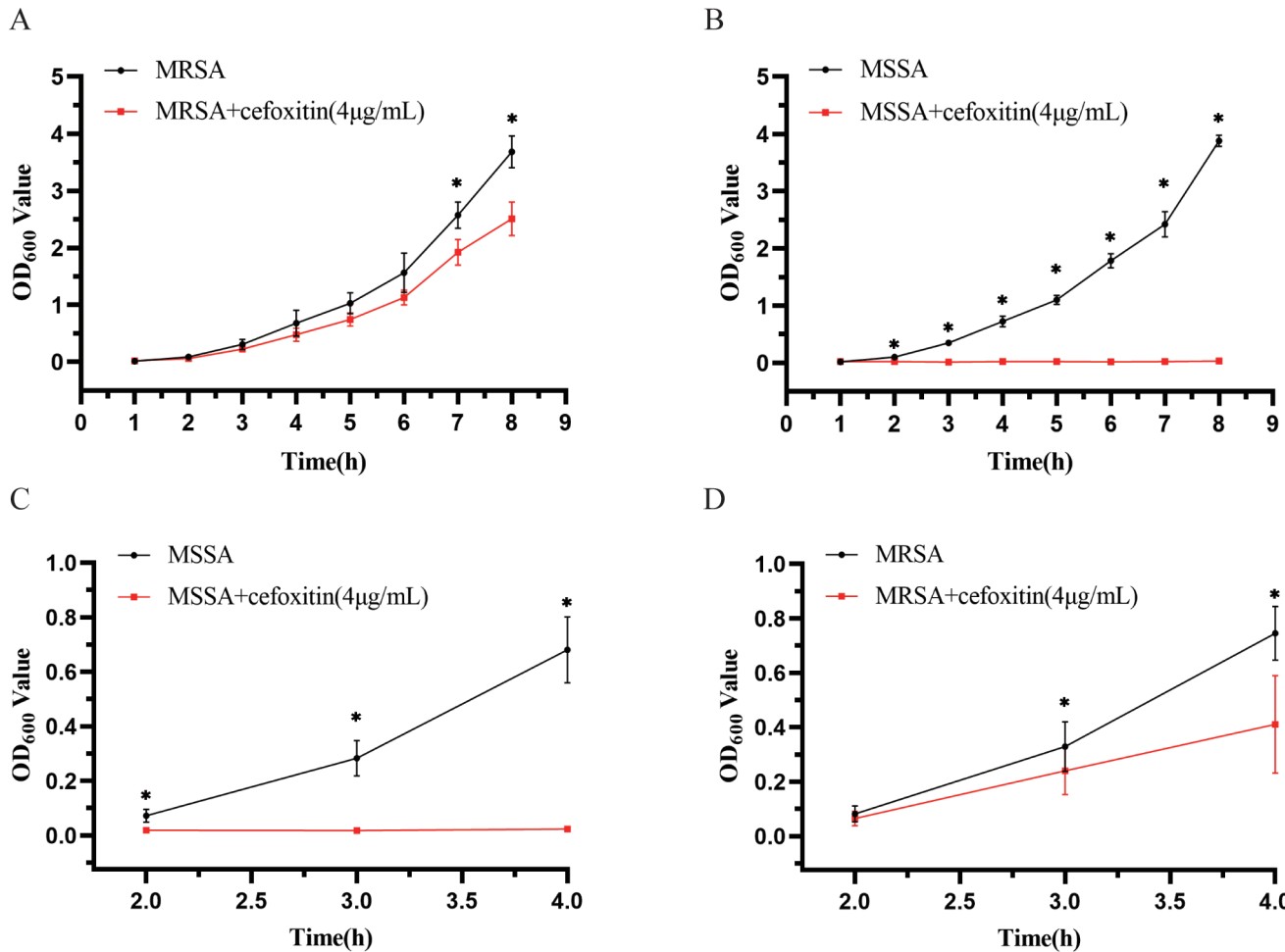

FIG 3 $OD_{600}$ values for MSSA and MRSA with culture time. (A) One strain of MRSA was cultured in CAMHB with or without cefoxitin (4 µg/mL) for 8 h. The experiment was repeated three times. (B) One strain of MSSA was cultured in CAMHB with or without cefoxitin (4 µg/mL) for 8 h. The experiment was repeated three times. (C) A total of 20 MRSA strains were cultured (2, 3, and 4 h) in CAMHB with or without cefoxitin (4 µg/mL). (D) A total of 20 MSSA strains were cultured (2, 3, and 4 h) in CAMHB with or without cefoxitin (4 µg/mL). $*P < 0.05$ between the control and drug groups.

and positive and negative predictive values of MRSA were all 100%, as identified by centrifugation-based enrichment after a 3-h culture. The results remained consistent with the aforementioned observations after a 4-h culture (Table S9). Limit of detection for *S. aureus* identification and methicillin resistance differentiation is shown in Table S11.

In clinical practice, the workflow begins with centrifuging the CSF (12,000 rpm, 5 min), inoculating 1 µL pellet into 9 µL LB broth for 8-h incubation at 35°C. The culture is then split: one portion is processed for MALDI-TOF MS identification of *S. aureus*, while the other is centrifuged and adjusted $OD_{600}$ to 0.7–0.8 with CAMHB, diluted 100-fold, and tested for methicillin resistance by mixing with cefoxitin (final 4 µg/mL), followed by 3-h incubation and MALDI-TOF MS analysis. Growth denotes MRSA; absence indicates MSSA, completing the process within 11 h (Fig. S2).

## Concordance analysis

The results of centrifugation-based enrichment combined with mass spectrometry and microbroth culture of 40 strains of *S. aureus* after a 3-h culture were analyzed by the Kappa test (Table S8). It was validated that the results of both methods were identical (Kappa = 1). These data refer to the identification of phenotypic oxacillin/methicillin resistance.

**TABLE 2** MRSA identification scores of different protocols in CHAMB (4 µg/mL cefoxitin)[a]

| Number | MIC (µg/mL) | Enrichment by non-centrifugation | | | Enrichment by centrifugation | | |
|---|---|---|---|---|---|---|---|
| | | 2 h | 3 h | 4 h | 2 h | 3 h | 4 h |
| R-1 | 8 | – | – | – | 2.22 ± 0.09 | 2.46 ± 0.15 | 2.35 ± 0.10 |
| R-2 | 32 | – | 2.25 ± 0.12 | 2.44 ± 0.18 | – | 2.36 ± 0.10 | 2.35 ± 0.08 |
| R-3 | 16 | – | – | 2.34 ± 0.15 | – | 2.46 ± 0.12 | 2.36 ± 0.09 |
| R-4 | 8 | – | – | – | 2.04 ± 0.08 | 2.53 ± 0.14 | 2.45 ± 0.11 |
| R-5 | 128 | – | 2.35 ± 0.13 | 2.45 ± 0.16 | – | 2.51 ± 0.10 | 2.29 ± 0.07 |
| R-6 | 64 | – | 2.10 ± 0.08 | 2.25 ± 0.11 | – | 2.28 ± 0.09 | 2.30 ± 0.06 |
| R-7 | 64 | – | 2.18 ± 0.10 | 2.44 ± 0.14 | – | 2.26 ± 0.07 | 2.35 ± 0.09 |
| R-8 | 32 | – | 2.28 ± 0.12 | 2.35 ± 0.10 | 2.46 ± 0.15 | 2.45 ± 0.13 | 2.34 ± 0.08 |
| R-9 | 8 | – | – | 2.09 ± 0.06 | – | 2.42 ± 0.11 | 2.49 ± 0.14 |
| R-10 | 32 | – | 2.39 ± 0.14 | 2.41 ± 0.12 | – | 2.40 ± 0.09 | 2.46 ± 0.10 |
| R-11 | 16 | – | 2.42 ± 0.16 | 2.36 ± 0.11 | 2.30 ± 0.08 | 2.26 ± 0.06 | 2.42 ± 0.12 |
| R-12 | 16 | – | 2.43 ± 0.13 | 2.41 ± 0.10 | 2.42 ± 0.09 | 2.32 ± 0.07 | 2.23 ± 0.05 |
| R-13 | 32 | – | 2.45 ± 0.18 | 2.30 ± 0.12 | 2.42 ± 0.10 | 2.38 ± 0.08 | 2.44 ± 0.11 |
| R-14 | 16 | – | – | 2.39 ± 0.13 | – | 2.33 ± 0.09 | 2.35 ± 0.07 |
| R-15 | 128 | – | 2.24 ± 0.10 | 2.28 ± 0.08 | – | 2.32 ± 0.06 | 2.40 ± 0.12 |
| R-16 | >256 | – | 2.25 ± 0.11 | 2.36 ± 0.09 | – | 2.20 ± 0.08 | 2.33 ± 0.07 |
| R-17 | 8 | – | – | – | – | 2.36 ± 0.12 | 2.41 ± 0.09 |
| R-18 | 64 | – | 2.44 ± 0.15 | 2.38 ± 0.11 | 2.39 ± 0.08 | 2.43 ± 0.10 | 2.42 ± 0.07 |
| R-19 | 8 | – | – | 2.44 ± 0.12 | 2.40 ± 0.09 | 2.28 ± 0.07 | 2.36 ± 0.08 |
| R-20 | 16 | – | 2.23 ± 0.10 | 2.35 ± 0.09 | – | 2.32 ± 0.07 | 2.37 ± 0.06 |
| | | 0 | 13 | 17 | 8 | 20 | 20 |

[a]The dash (–) indicates that the bacteria have not been conclusively identified.

## DISCUSSION

As a severe infectious disease involving the central nervous system (CNS), bacterial meningitis commonly affects adults and is particularly prevalent in children, with a high mortality rate and accompanied by an increased incidence of sequelae (21, 22). Infection with *S. aureus* is one of the causes of bacterial meningitis, presenting a particular challenge in its management and treatment (23). A timely and effective antibiotic treatment is considered essential for improving the cure rate of this disease. However, the rapid development of antibiotic resistance, particularly with MRSA, poses a significant challenge to the cure of this disease (24–26). The rapid identification of MRSA is considered beneficial to the early treatment of patients, thus reducing the risk of death (16, 27). However, potential contamination conditions may impact bacterial identification.

In this study, the enrichment efficiency of bacteria was evaluated. The results demonstrated that the efficiency of the commonly used centrifugation method can reach over 85%, highlighting the simplicity and reliability of this method. In terms of bacterial identification, it was found that a rapid broth culture of 7 or 8 h was sufficient to meet the requirements for mass spectrometry detection, significantly reducing the time required by conventional methods. For the identification of antibiotic resistance, the differentiation between MRSA and MSSA was made based on their distinct sensitivity to antibiotics. It was observed that MSSA could not grow in the broth culture media containing cefoxitin, and none of the 20 sensitive strains were successfully identified. For MRSA, 12 strains of *S. aureus* were not successfully identified after a 2-h culture, but the successful identification of these strains was achieved after a 3- and 4-h culture. Therefore, 3 h can be considered the optimal culture time for MRSA identification. In comparison, the optimal culture time of the DOT-MGA method used by Idelevich et al. to identify MRSA was determined to be 5 h (16). In this study, the detection of the $OD_{600}$ value of bacterial suspensions can be achieved through ultraviolet-visible spectroscopy (UV-VIS). In CAMHB with cefoxitin, the changes in the $OD_{600}$ value appeared after a 2-h

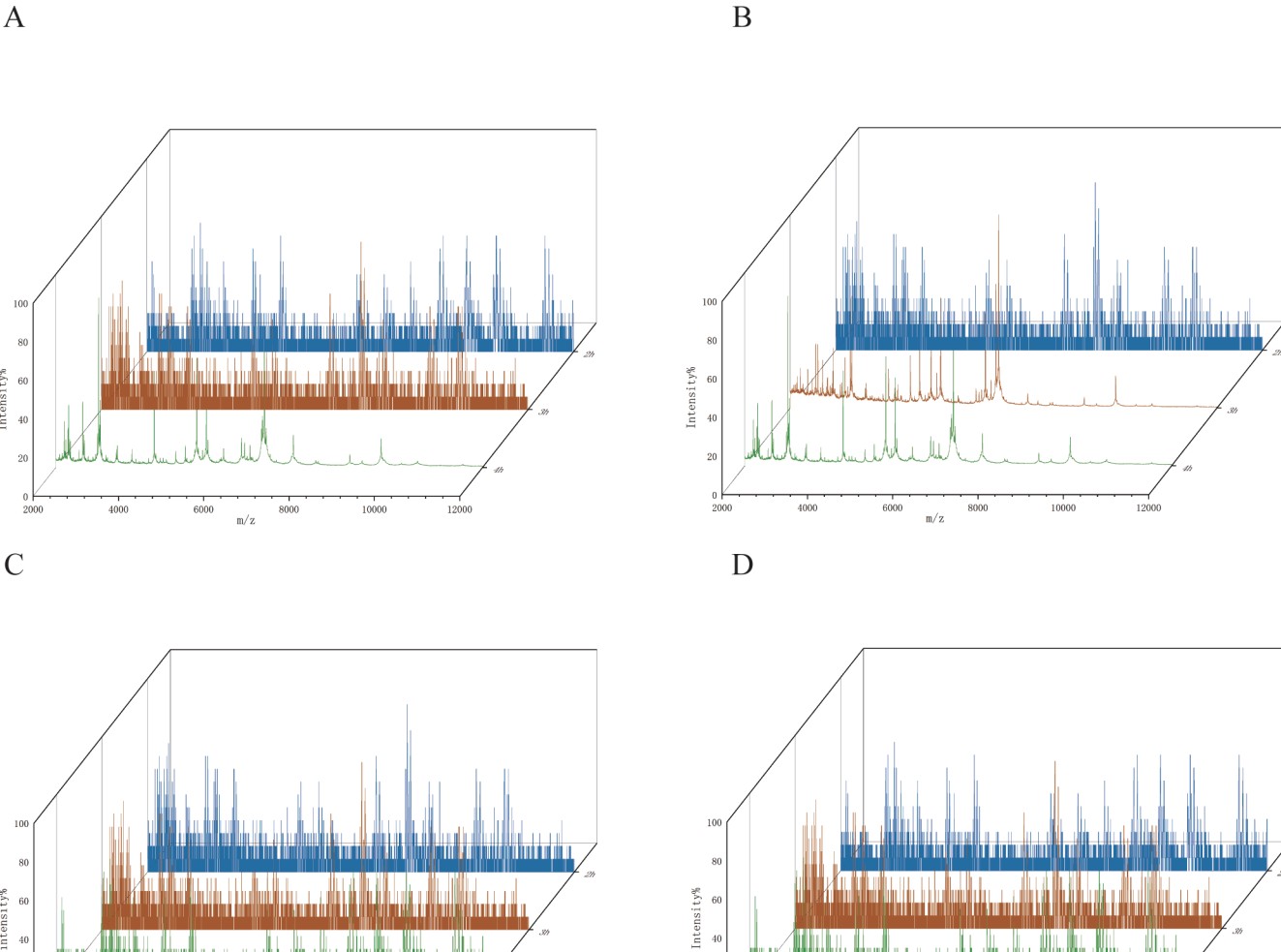

**FIG 4** MALDI-TOF MS spectra of clinical *S. aureus* strains, including 2 (blue), 3 (orange), and 4 h (green). (A) MRSA+ enrichment by non-centrifugation. (B) MRSA+ enrichment by centrifugation. (C) MSSA+ enrichment by non-centrifugation. (D) MSSA+ enrichment by centrifugation.

culture, and the lower absorbance at this stage was correlated with a lower identification rate. A significant change in the $OD_{600}$ value was observed after a 3-h culture, marking this time length as the optimal culture time for MRSA identification. As a result, the $OD_{600}$ value can serve as an auxiliary method to ascertain the necessity for the enrichment medium and mass spectrometry.

While this study demonstrates the efficacy of MALDI-TOF MS for rapid *S. aureus* identification and methicillin resistance testing in CSF, several limitations should be

**TABLE 3** The resistance of *S. aureus* to methicillin was identified by MALDI-TOF MS in the cerebrospinal fluid at 3 h

| Evaluation index | Enrichment by non-centrifugation | Enrichment by centrifugation |
|---|---|---|
| Validity (%) | 82.5 | 100 |
| Sensitivity (%) | 65 | 100 |
| Specificity (%) | 100 | 100 |
| Positive predictive value (%) | 100 | 100 |
| Negative predictive value (%) | 74.1 | 100 |

acknowledged. First, the focus was exclusively on *S. aureus*, which limits applicability to other meningitis pathogens, such as *S. pneumoniae* and *N. meningitidis*; second, the optimized workflow parameters were established empirically in this study and may require further standardization for broader adoption. Future studies should expand pathogen coverage and test clinical samples via multicenter collaboration.

## Conclusion

This report offers a mass spectrometry-based method that utilized centrifugal enrichment for the rapid identification of pathogens and their resistance, and the identification and resistance analysis of *S. aureus* in the cerebrospinal fluid (CSF) were completed within 11 hours. In this study, the CSF was used to simulate the environment by adding bacteria, thus achieving clinical application. Nevertheless, the results should be validated in a study based on a larger sample size. Moreover, it is necessary to standardize and optimize testing conditions and evaluation criteria in further research. It is expected that this method will guide clinicians in the timely and correct application of antibiotics after the validation of various resistant bacteria.

## ACKNOWLEDGMENTS

All authors made substantial contributions to the conception and design, acquisition of data, or analysis and interpretation of data, took part in drafting the article or revising it critically for important intellectual content, agreed to submit it to the current journal, gave final approval for the version to be published, and agreed to be accountable for all aspects of the work.

This work was supported by Suzhou Municipal Health Commission Project (SZWJ2023a056).

Informed consent has been obtained from the patient.

## AUTHOR AFFILIATIONS

[1]Wanbei Coal Electricity Group General Hospital, Suzhou, China
[2]The First Affiliated Hospital of Anhui Medical University, Hefei, China
[3]TongLing People's Hospital, Tongling, Anhui, China

## AUTHOR ORCIDs

Mengyu Zhang  http://orcid.org/0009-0005-4853-4532
Kaixuan Zhang  http://orcid.org/0009-0004-7521-5062
Sudi Zhu  http://orcid.org/0000-0001-5522-7382
Yuanyuan Xu  http://orcid.org/0009-0002-6340-2632

## FUNDING

| Funder | Grant(s) | Author(s) |
| --- | --- | --- |
| Suzhou Municipal Health Commission | SZWJ2023a056 | Ailing Ma |
| | | Shuguo Qin |
| | | Di Hu |
| | | Henggui Hu |
| | | Xiaolei Du |
| | | Kaixuan Zhang |
| | | Sudi Zhu |
| | | Yuanyuan Xu |

## AUTHOR CONTRIBUTIONS

Mengyu Zhang, Funding acquisition, Investigation, Methodology, Validation, Visualization, Writing – original draft, Writing – review and editing | Xuanxuan Wang, Data curation, Validation, Visualization, Writing – original draft, Writing – review and editing | Wei Huang, Formal analysis | Ailing Ma, Conceptualization | Shuguo Qin, Project administration, Validation, Visualization | Di Hu, Data curation, Funding acquisition, Investigation | Henggui Hu, Data curation, Methodology | Xiaolei Du, Methodology, Project administration | Kaixuan Zhang, Resources, Supervision, Validation, Visualization | Sudi Zhu, Methodology, Software, Validation, Visualization | Yuanyuan Xu, Methodology, Project administration, Resources, Software, Supervision, Validation, Visualization

## DATA AVAILABILITY

Data are provided within the manuscript or Supplemental material. The MALDI-TOF MS data sets for the current study are available from the corresponding author on reasonable request.

## ETHICS APPROVAL

This study was approved by the Ethics Committee of the Wanbei Coal Electric Group General Hospital (WBZY-LLWYH-2025-023) in accordance with the Declaration of Helsinki. In addition, the study obtained informed consent from all subjects.

## ADDITIONAL FILES

The following material is available online.

### Supplemental Material

**Supplemental material (Spectrum01508-25-S0001.docx).** Fig. S1 and S2; Tables S1 to S11.

### Open Peer Review

**PEER REVIEW HISTORY (review-history.pdf).** An accounting of the reviewer comments and feedback.

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
