## [Reviewer comments · Microbiology Spectrum]

Microbiology Spectrum

Rapid Identification and Methicillin Resistance Test to *Staphylococcus aureus* in Cerebrospinal by MALDI-TOF MS

Mengyu Zhang, Xuanxuan Wang, Wei Huang, Ailing Ma, Shuguo Qin, Di Hu, Henggui Hu, Xiaolei Du, Kaixuan Zhang, Sudi Zhu, and Yuanyuan Xu

Corresponding Author(s): Mengyu Zhang, Anhui Wanbei Coal Electricity Group General Hospital

Review Timeline:

Submission Date:	May 15, 2025
Editorial Decision:	July 5, 2025
Revision Received:	August 25, 2025
Editorial Decision:	December 8, 2025
Revision Received:	January 9, 2026
Accepted:	February 22, 2026

Editor: Rebecca Yee

Reviewer(s): The reviewers have opted to remain anonymous.

Transaction Report:

DOI: <https://doi.org/10.1128/spectrum.01508-25>

Re: Spectrum01508-25 (**Rapid Identification and Methicillin Resistance Test to *Staphylococcus aureus* in Cerebrospinal by MALDI-TOF MS**)

Dear Mr. Mengyu Zhang:

Thank you for the privilege of reviewing your work. Below you will find my comments, instructions from the Spectrum editorial office, and the reviewer comments.

One major issue with the manuscript is that it appears to overstate several of its claims. The study claims to detect pathogens directly from CSF but only focuses on *S. aureus*, with unclear methods regarding CSF use and no evidence of testing actual patient samples with *S. aureus*, leading to concerns about consistency and clinical relevance. There is no discussion regarding the clinical feasibility of this approach, and key methodological details are missing.

Revision Guidelines

Sincerely,
Rebecca Yee
Editor
Microbiology Spectrum

Reviewer #1 (Comments for the Author):

Summary of Key Findings:

As bacterial meningitis can lead to high morbidity and mortality, the authors discuss the benefits of early diagnosis and application of appropriate antimicrobial treatment for this disease. As a potential cause of bacterial meningitis, the authors investigate diagnostic methodologies to promote early identification of *Staphylococcus aureus* from cerebrospinal fluid (CSF) and differentiation of methicillin resistant *S. aureus* (MRSA) and methicillin sensitive *S. aureus* (MSSA). Experiments assessed the following key methodology steps to assist in rapid *S. aureus* identification from CSF: 1) short term rapid culture, 2) centrifugation enrichment after short term rapid culture, 3) MALDI-TOF mass spectrometry identification of enriched precipitate obtained by centrifugation procedure. Experiments assessed the following key methodology steps to assist in rapid *S. aureus* differentiation of phenotypic methicillin resistance or methicillin sensitivity after initial *S. aureus* species identification: 1) short term rapid culture with cation-adjusted Mueller-Hinton broth (CAMHB) supplemented with cefoxitin (4 µg/mL), 2) centrifugation enrichment after short term rapid culture, 3) MALDI-TOF mass spectrometry performance on enriched precipitate obtained by centrifugation procedure. The data provided in this rapid testing evaluation are interesting to review from a clinical diagnostics perspective, yet aspects of the manuscript require further expansion in the text. Presumably, optimal parameters (culture volumes, culture time, controls, etc.) of the proposed diagnostic testing for CSF *S. aureus* was determined by the experimental data, yet it is not straightforward for the reader to put together the entire proposed diagnostic workflow and see how this could be utilized in a clinical setting. Additionally, no limit of detection (LOD) studies were performed. Please see the major and minor comments proposed below.

Major Concerns:

- 1) Synthesized from the experimental results, a summary paragraph (supplemented with a figure) is recommended to lay out how the authors envision this CSF *S. aureus* diagnostic workflow to be utilized in a true diagnostic setting, e.g., controls required, seeding specimen volumes, culture media diluent volumes, rapid culture duration, culture optical densities required for downstream procedures, parallel or sequential *Staphylococcus* species identification and MRSA/MSSA differentiation testing, etc. Both arms (Staph species ID vs. MRSA/MSSA differentiation) of diagnostic testing should be clarified. Because the listed experiments utilized many moving parts, including preparation of bacterial stock solutions and subsequent plating and counting of organisms to aid in their quantitation, as well as appropriate controls, it is challenging for the reader to put together and envision the entire diagnostic workflow that would occur and could be completed in the proposed 11-hour turnaround time.
- 2) 'Bacterial strains' section in 'Materials and Methods': It is recommended to describe what sources the 40 clinical *S. aureus* isolates (unique patients) were obtained from for experimental testing (i.e., were they all isolated from CSF from cases of bacterial meningitis, or a mix of other anatomical locations associated with other types of *S. aureus* infections?); Line 30: It is stated that the *mecA* gene was detected in all 40 MRSA isolates used in the experiments. Please describe the methodology used allowing for determination of *mecA* gene presence.
- 3) 'MIC determination' section in 'Materials and Methods': it is stated that broth microdilution was used; was there a specific commercial platform that was used or was gold standard testing performed (preparation of in-house antibiotic dilutions and antibiotic panel preparation, etc.)? This should be specified. Lines 143-144: the specific CLSI document(s) and ISO guideline document(s) that were used to guide this testing should both be included and cited in-text and in the references list correctly.
- 4) Since this manuscript focuses on rapid identification methods of *S. aureus* (MRSA and MSSA), for clinical context, it would be useful to describe the rates of bacterial meningitis that are caused by *S. aureus* potentially from both a global perspective, as well as from the local healthcare institutions from where the authors practice. This information would help to set up the clinical need and clinical utility of the proposed diagnostic testing.
- 5) Limit of detection studies should be performed for both arms of the proposed diagnostic testing workflows (*Staphylococcus* species ID and MRSA/MSSA differentiation).
- 6) A paragraph describing limitations should be included in the discussion.

Minor Concerns:

1. Material and Methods: it is suggested to avoid repeating the same items if they are unnecessary. Key methods can be defined in the material and methods section only, whereas in the results section, the reader can be referred to the previously defined materials/methods.
2. Proofing: 1) apply superscripts correctly in all defined CFU/mL numbers, 2) ensure 'cation-adjusted Mueller-Hinton broth (CAMHB)' is abbreviated correctly throughout the document, 3) define key data values being displayed (e.g., mean/median with standard deviation/range, define 'P' as 'p-value', etc.), 4) Line 300: change to "Concordance analysis" or "Result agreement" analysis; further specify these data refer to identification of phenotypic oxacillin/methicillin resistance; 5) Table S3: recheck values, i.e., *S. aureus* (40) and *S. aureus* (160) at 2 hours percentages appear incorrect, 6) Abbreviations section: 'DOT-MGA' appears to stand for 'Direct-on-Target Microdroplet Growth Assay'.
3. To further define: 1) MALDI-TOF MS instrumentation/system used for experiments, 2) Materials and Methods section: specification of specimen/culture incubation conditions (temperature, humidity, CO₂ percentage if included, shaking-if applicable, etc.), 3) line 149: specify specific 'nutrition agar plates' used.
4. It would be useful for the supplemental figures/tables to have their own brief legend.
5. Figure 2: it may be useful to include a specific color in the legend to designate what is meant to be CSF. In the materials and methods section, it would be useful to specify what type of CSF specimens were used in this study. For example, were all CSF specimens in this study obtained by lumbar puncture only or were other types of CSF collections also tested?
6. It is suggested to describe how enrichment efficiency was calculated.
7. Lines 246-247: it is suggested to specify why OD₆₀₀ of >0.8 was best or optimal for bacterial identification. There is later description of data that show that OD₆₀₀ of <0.8 is adequate for successful MALDI-TOF MS identification. Please provide further clarity on this.

8. Table 3: it is suggested to revise this table to show appropriate contingency tables for each method group and time point (2, 3, and 4 hours). It could be considered to display contingency tables of only the selected optimal culture time point in the manuscript, while the contingency tables of the other (less optimal) culture time points could be displayed as a supplemental table.

9. Table S1: Please clarify if the 1 microliter of specimen added to the LB or CSF diluent is the original CSF specimen with *S. aureus* incorporated in the specimen. Please describe what the CSF diluent is-e.g., is this a confirmed bacterial culture-negative CSF specimen or pooled specimens used as a 'growth control'?

Reviewer #2 (Comments for the Author):

In this manuscript, the authors develop a proof of concept method utilizing growth kinetics, centrifugation, and MALDI-TOF MS to identify *S. aureus* strains. The authors demonstrate that incubation in media containing cefoxitin results in growth of MRSA strains but not MSSA strains, and hypothesize that this method could be extrapolated to clinical samples such as CSF. Although an interesting concept, there are many issues with the manuscript methods and details that need to be addressed.

Major comments

1. The authors report in the manuscript that rapid pathogen identification and drug resistance identification can occur using this method. However, in this manuscript, the authors only describe *S. aureus* and MSSA/MRSA. The language should be adjusted throughout to focus on this organism.
2. In the introduction, the authors state that *S. aureus* is a cause of meningitis. While this is true, it is a very minor cause (1-3%) often due to other underlying diseases. Please update to reflect this. The authors should state why they chose *S. aureus* for this study.
3. In the introduction, the authors state that maldi-tof ms has been used for detection of drug resistance in bacteria. While true, these are research concepts and are not actually employed in clinical labs. The authors should adjust their language to reflect this.
4. In the methods, line 132 states that micro broth dilution was used but no details are given on what platform or what standards of interpretation are used.
5. Lines 167-205, it is confusing where CSF is involved in this study. It appears in Figure 2 as if CSF containing organisms is being directly inoculated with LB broth for growth determination and resistance detected, but that is not what is described in the methods. Rather, CSF was used as a growth control to show that *S. aureus* does not grow in CSF alone but grew in LB and CAMHB. There are no patient samples of CSF grown to show that this concept applies directly to csf specimens. If CSF is used, then the ethics statement must be updated to reflect use of patient samples.
6. How applicable is this clinically? Are labs expected to add LB or CAMHB to a csf sample? If this takes 8 hours to grow and work, how different is this than utilizing a rapid molecular test, identification by culture the next day, and a rapid enzyme assay to identify PBP2A?

Minor comments

1. In the methods, it is stated that *mecA* was detected in all MRSA isolates but it does not list the method for how that was performed.
2. Lines 142-144, a reference is given for CLSI but it is very outdated. ISO is also mentioned but no reference is given.
3. Line 177, what maldi instrument is used and which database?
4. Line 254, what is the range expected for the QC organism and by what standards?

Dear professor,

On behalf of my co-authors, we thank you very much for giving us an opportunity to revise our manuscript, and we appreciate the editor and reviewer very much for their thoughtful and constructive comments and suggestions on our manuscript entitled "Rapid Identification and Methicillin Resistance Test to *Staphylococcus aureus* in Cerebrospinal by MALDI-TOF MS". We have tried our best to revise our manuscript according to the comments. Our point-by-point responses to the reviewers' comments are detailed.

Sincerely Yours,

Mengyu Zhang, Yuanyuan Xu

Wanbei Coal Electric Group General Hospital, Suzhou 234011, China

Reviewer #1

#1. Synthesized from the experimental results, a summary paragraph (supplemented with a figure) is recommended to lay out how the authors envision this CSF *S. aureus* diagnostic workflow to be utilized in a true diagnostic setting, e.g., controls required, seeding specimen volumes, culture media diluent volumes, rapid culture duration, culture optical densities required for downstream procedures, parallel or sequential *Staphylococcus* species identification and MRSA/MSSA differentiation testing, etc. Both arms (*Staph* species ID vs. MRSA/MSSA differentiation) of diagnostic testing should be clarified. Because the listed experiments utilized many moving parts, including preparation of bacterial stock solutions and subsequent plating and counting of organisms to aid in their quantitation, as well as appropriate controls, it is challenging for the reader to put together and envision the entire diagnostic workflow that would occur and could be completed in the proposed 11-hour turnaround time.

The authors' answer: Thank you for your comment. We have added a dedicated summary paragraph to explicitly outline the envisioned clinical workflow. (Page 8, lines 325-331; Figure S2)

#2. 'Bacterial strains' section in 'Materials and Methods': It is recommended to describe what sources the 40 clinical *S. aureus* isolates (unique patients) were obtained from for experimental testing (i.e., were they all isolated from CSF from cases of bacterial meningitis, or a mix of other anatomical locations associated with other types of *S. aureus* infections?); Line 30: It is stated that the *mecA* gene was detected in all 40 MRSA isolates used in the experiments. Please describe the methodology used allowing for determination of *mecA* gene presence.

The authors' answer: I agree with your comment and thank you for your suggestion. We have revised the "Bacterial strains", explicitly stating that isolates were obtained from CSF of patients with bacterial meningitis and other clinical sites (sputum, mucosal secretions, and cutaneous secretions). This modification better reflects the clinical context of the strains and enhances the generalizability of our findings. We have updated the Methods section to specify the *mecA* detection method as requested. The *mecA* was detected via PCR using species-specific primers (*mecA*-F/R), with reaction conditions including 35 cycles of denaturation (95°C, 30s), annealing (58°C, 30s), and extension (72°C, 30s). Amplicons were verified by 1% agarose gel

electrophoresis and Sanger sequencing (Sangon Biotech, China), with sequences aligned to NCBI GenBank for confirmation. (Page 4, lines138-140; lines141-146; Table S10)

#3. 'MIC determination' section in 'Materials and Methods': it is stated that broth microdilution was used; was there a specific commercial platform that was used or was gold standard testing performed (preparation of in-house antibiotic dilutions and antibiotic panel preparation, etc.)? This should be specified. Lines 143-144: the specific CLSI document(s) and ISO guideline document(s) that were used to guide this testing should both be included and cited in-text and in the references list correctly.

The authors' answer: Thank you for your advice. We confirm that the MIC testing was conducted using the standard broth microdilution method with in-house prepared antibiotic panels, adhering strictly to CLSI and ISO recommendations for manual dilution preparation and quality control. The specific CLSI and ISO standard have been added to the manuscript. (Reference 19 and 20). (Page 4, lines 151-154)

#4. Since this manuscript focuses on rapid identification methods of *S. aureus* (MRSA and MSSA), for clinical context, it would be useful to describe the rates of bacterial meningitis that are caused by *S. aureus* potentially from both a global perspective, as well as from the local healthcare institutions from where the authors practice. This information would help to set up the clinical need and clinical utility of the proposed diagnostic testing.

The authors' answer: I agree with your comment and thank you for your suggestion. We have added the data on the global and local incidence rates of *S. aureus* bacterial meningitis in the revised manuscript. (Page 3, Lines 96-100)

#5. Limit of detection studies should be performed for both arms of the proposed diagnostic testing workflows (Staphylococcus species ID and MRSA/MSSA differentiation).

The authors' answer: Thank you for your advice. We have performed the limit of detection studies for both arms of the proposed diagnostic testing workflows in the manuscript. (Page 8, Lines 324-325; Table S11)

#6. A paragraph describing limitations should be included in the discussion.

The authors' answer: Thank you for your comment. We have added a dedicated limitations paragraph in the discussion. (Page 9, Lines 373-380)

Minor Concerns:

#1. Material and Methods: it is suggested to avoid repeating the same items if they are unnecessary. Key methods can be defined in the material and methods section only, whereas in the results section, the reader can be referred to the previously defined materials/methods.

The authors' answer: Thank you for your advice. We have removed redundant methodological descriptions in the results section, referring readers to the material and methods section for key protocols. (Page 7, Lines 265-266)

#2. Proofing: 1) apply superscripts correctly in all defined CFU/mL numbers, 2) ensure

'cation-adjusted Mueller-Hinton broth (CAMHB)' is abbreviated correctly throughout the document, 3) define key data values being displayed (e.g., mean/median with standard deviation/range, define 'P' as 'p-value', etc.), 4) Line 300: change to "Concordance analysis" or "Result agreement" analysis; further specify these data refer to identification of phenotypic oxacillin/methicillin resistance; 5) Table S3: recheck values, i.e., *S. aureus* (40) and *S. aureus* (160) at 2 hours percentages appear incorrect, 6) Abbreviations section: 'DOT-MGA' appears to stand for 'Direct-on-Target Microdroplet Growth Assay'.

The authors' answer: Thank you for your advice. We have corrected all CFU/mL numbers to proper superscript formatting, ensured consistent abbreviation of "cation-adjusted Mueller-Hinton broth (CAMHB)" after its first definition, and explicitly defined key statistical values including mean with standard deviation as well as clarifying "P" as p-value in the manuscript. We have revised Line 300 to "Concordance analysis", specified these data refer to identification of phenotypic oxacillin/methicillin resistance, rechecked and corrected values in Table S3, and confirmed "DOT-MGA" as "Direct-on-Target Microdroplet Growth Assay" in the abbreviations section. All these modifications have been incorporated into the revised manuscript,(Page 2, Lines 56-61; Page 5, Lines 201-211; Page 8, Lines 311-316; Table 1; Table 2; Table S3)

#3. To further define: 1) MALDI-TOF MS instrumentation/system used for experiments, 2) Materials and Methods section: specification of specimen/culture incubation conditions (temperature, humidity, CO₂ percentage if included, shaking-if applicable, etc.), 3) line 149: specify specific 'nutrition agar plates' used.

The authors' answer: Thank you for your comment. We have clarified the MALDI-TOF MS system details in manuscript. We have updated the Materials and Methods section to specify that all liquid cultures were incubated at 37°C under aerobic conditions with continuous orbital shaking at 900 rpm. We have clarified the brand and source as "nutrient agar plates (Tianda Reagent, China). (Page 4, Lines 150-151; Lines 167-168; Page 6, Lines 232-237)

#4. It would be useful for the supplemental figures/tables to have their own brief legend.

The authors' answer: Thank you for your suggestion. We have now added concise descriptive legends to all supplemental figures and tables to improve clarity and accessibility.

#5. Figure 2: it may be useful to include a specific color in the legend to designate what is meant to be CSF. In the materials and methods section, it would be useful to specify what type of CSF specimens were used in this study. For example, were all CSF specimens in this study obtained by lumbar puncture only or were other types of CSF collections also tested?

The authors' answer: Thank you for your advice. We have specified in Materials and Methods that all CSF specimens were obtained exclusively via lumbar puncture with standardized processing protocols.(Page 4, Lines 150-153; Figure 2)

#6. It is suggested to describe how enrichment efficiency was calculated.

The authors' answer: Thank you for your comment. We have described the calculation method for enrichment efficiency in the manuscript.(Page 5, Lines 181-183)

#7. Lines 246-247: it is suggested to specify why OD₆₀₀ of >0.8 was best or optimal for bacterial

identification. There is later description of data that show that OD₆₀₀ of <0.8 is adequate for successful MALDI-TOF MS identification. Please provide further clarity on this.

The authors' answer: Thank you for your advice. We standardized the OD₆₀₀ threshold at 0.7 – 0.8 because this range corresponds to ~10⁸ CFU/mL bacterial density, enabling direct sampling of 1 µL (~10⁵ CFU) for MALDI-TOF MS analysis, which aligns with the minimum detection limit reported by Ying et al¹. However, subsequent experiments demonstrated that even lower OD₆₀₀ values (<0.8) could still successful identification through centrifugal enrichment of bacterial pellets, a strategy validated by both methodological consistency and experimental data.

#8. Table 3: it is suggested to revise this table to show appropriate contingency tables for each method group and time point (2, 3, and 4 hours). It could be considered to display contingency tables of only the selected optimal culture time point in the manuscript, while the contingency tables of the other (less optimal) culture time points could be displayed as a supplemental table.

The authors' answer: Thank you for your advice. The Table 3 has been revised accordingly, with remaining data presented in supplementary Table S9. (Page 8, Lines 323; Table S9)

#9. Table S1: Please clarify if the 1 microliter of specimen added to the LB or CSF diluent is the original CSF specimen with *S. aureus* incorporated in the specimen. Please describe what the CSF diluent is—e.g., is this a confirmed bacterial culture-negative CSF specimen or pooled specimens used as a 'growth control'?

The authors' answer: I agree with your comment and thank you for your suggestion. We have revised the table to clarify that the 1 µL specimen added to the LB or CSF diluent is the original CSF specimen with *S. aureus* incorporated in the specimen, and the CSF diluent is a confirmed bacterial culture-negative CSF specimen. These updates are now included in the supplement material. (Table S1)

Reviewer #2

Major comments

#1. The authors report in the manuscript that rapid pathogen identification and drug resistance identification can occur using this method. However, in this manuscript, the authors only describe *S. aureus* and MSSA/MRSA. The language should be adjusted throughout to focus on this organism.

The authors' answer: Thank you for your advice. We have revised the manuscript to ensure the descriptions focus specifically on *S. aureus* and MSSA/MRSA, aligning the language with the research scope. (Page 2, Lines 50-72; Page 3, Lines 129; Page 5, Lines 187; Page 6, Lines 262; Table 1.)

#2. In the introduction, the authors state that *S. aureus* is a cause of meningitis. While this is true, it is a very minor cause (1-3%) often due to other underlying diseases. Please update to reflect this. The authors should state why they chose *S. aureus* for this study.

The authors' answer: I agree with your comment and thank you for your suggestion. *S. aureus* is a relatively minor cause of bacterial meningitis, however, its clinical significance persists due to antibiotic resistance (notably MRSA) and high mortality rates (14-56%). At our institution (Wanbei Coal and Electricity Group General Hospital, Anhui, China), empirical data indicate that

S. aureus is the primary Gram-positive pathogen identified in meningitis diagnoses. Given its global and local therapeutic challenges, we selected *S. aureus* as the focus of this study. (Page 3, Lines 96-100)

#3. In the introduction, the authors state that maldi-tof ms has been used for detection of drug resistance in bacteria. While true, these are research concepts and are not actually employed in clinical labs. The authors should adjust their language to reflect this.

The authors' answer: Thank you for your comment. We have revised the text to clarify that MALDI-TOF MS is currently under investigation for bacterial drug resistance detection, rather than being routinely used in clinical practice." (Page 3, Lines 118-120)

#4. In the methods, line 132 states that micro broth dilution was used but no details are given on what platform or what standards of interpretation are used.

The authors' answer: Thank you for your advice. The micro broth dilution was performed in accordance with the CLSI and ISO reference methods. All antibiotic dilutions and susceptibility panels were prepared in-house following these standardized guidelines. (Page 4, lines 160-162)

#5. Lines 167-205, it is confusing where CSF is involved in this study. It appears in Figure 2 as if CSF containing organisms is being directly inoculated with LB broth for growth determination and resistance detected, but that is not what is described in the methods. Rather, CSF was used as a growth control to show that *S aureus* does not grow in CSF alone but grew in LB and CAMHB. There are no patient samples of CSF grown to show that this concept applies directly to csf specimens. If CSF is used, then the ethics statement must be updated to reflect use of patient samples.

The authors' answer: I agree with your comment and thank you for your suggestion. We have been revised to accurately reflect the experimental procedure in Figure 2: CSF was mixed with bacteria to emulate clinical infection samples, which was then centrifuged, and the resulting pellet was inoculated into LB broth for rapid growth determination and resistance detection. This modification aligns the figure with the detailed methods described in the manuscript. We have updated the ethical statement in the manuscript. (Page 10, lines 419-422; Figure 2).

#6. How applicable is this clinically? Are labs expected to add LB or CAMHB to a csf sample? If this takes 8 hours to grow and work, how different is this than utilizing a rapid molecular test, identification by culture the next day, and a rapid enzyme assay to identify PBP2A?

The authors' answer: Thank you for your advice. This method shows high clinical potential for rapid MRSA detection in CSF (11-hour turnaround, 100% accuracy), but requires validation with larger sample size and broader pathogen coverage. Labs actually need to centrifuge patient CSF sample to enrich bacteria, then add the resulting pellet to LB or CAMHB for rapid cultivation and antimicrobial susceptibility testing. The rapid molecular testing and other methods, while capable of providing accurate information, require complex professional equipment and high costs. In contrast, our proposed method is simpler to perform, more cost-effective, rapidly meets clinical needs for preliminary resistance assessment, reduces sample processing steps and wait times, and better aligns with the practical scenario of limited medical resources in primary care settings.

Minor comments

#1. In the methods, it is stated that *mecA* was detected in all MRSA isolates but it does not list the method for how that was performed.

The authors' answer: Thank you for your comment. We have supplemented the Methods section to specify: *mecA* was detected via PCR using species-specific primers (*mecA*-F/R), with reaction conditions including 35 cycles of denaturation (95°C, 30s), annealing (58°C, 30s), and extension (72°C, 30s). Amplicons were verified by 1% agarose gel electrophoresis and Sanger sequencing (Sangon Biotech, China), with sequences aligned to NCBI GenBank for confirmation. This ensures robust and reproducible detection. Thank you for your input, which strengthens methodological transparency. (Page 4, lines 141-146)

#2. Lines 142-144, a reference is given for CLSI but it is very outdated. ISO is also mentioned but no reference is given.

The authors' answer: Thank you for your advice. We have updated the manuscript with the most recent CLSI guidelines. Reference to CLSI and ISO are added in the manuscript. (Page 4, lines 161-162; Reference 19 and 20)

#3. Line 177, what maldi instrument is used and which database?

The authors' answer: Thank you for your comment. We have specified the maldi instrument as the MALDI Biotyper (Bruker Daltonics, Germany) with the MBT RUO-DB8468 database in the revised manuscript (Page 6, lines 232-237)

#4. Line 254, what is the range expected for the QC organism and by what standards?

The authors' answer: Thank you for your comment. The expected QC organism range is 1-4 µg/mL, which is based on the standards outlined in CLSI M100 (Performance Standards for Antimicrobial Susceptibility Testing). (Page 7, lines 278-280)

Re: Spectrum01508-25R1 (**Rapid Identification and Methicillin Resistance Test to *Staphylococcus aureus* in Cerebrospinal by MALDI-TOF MS**)

Dear Mr. Mengyu Zhang:

Thank you for the privilege of reviewing your work. Below you will find my comments and instructions from the Spectrum editorial office.

Upon review with the editorial board, there were major concerns regarding the methods described by the authors as they are not those described by ISO or CLSI standards. If using OD to measure equivalent of 0.5 McFarland, the standards state the OD at 625 nm wavelength should have reading of 0.08-0.13. Also, incubation temp of 37 rather than 35 C is high and may not detect methicillin resistance per CLSI M100. We suggest that the authors provide better clarification. If CLSI or ISO standards were not followed, then these documents should not be cited. Please provide other data regarding QC (or correlation between phenotypic and molecular AST results) and a more appropriate citation to justify the parameters that were used. The limitations regarding the method would need to be greatly expanded.

Revision Guidelines

Sincerely,
Rebecca Yee
Editor
Microbiology Spectrum

Dear professor,

On behalf of my co-authors, we would like to express our deepest gratitude for giving us another opportunity to further revise our manuscript, and we sincerely appreciate the editor and reviewers for their thoughtful and constructive comments provided on our manuscript entitled “Rapid Identification and Methicillin Resistance Test to *Staphylococcus aureus* in Cerebrospinal by MALDI-TOF MS”. We have diligently revised the manuscript in line with the feedback, and a detailed point-by-point response is provided below.

Sincerely Yours,

Mengyu Zhang, Yuanyuan Xu

Wanbei Coal Electric Group General Hospital, Suzhou 234011, China

#1. If using OD to measure equivalent of 0.5 McFarland, the standards state the OD at 625 nm wavelength should have reading of 0.08-0.13.

The authors’ answer: Thank you to the reviewer for highlighting the CLSI guideline regarding the use of OD₆₂₅ for turbidity adjustment. We fully understand and respect this standard. In our study, we chose OD₆₀₀ as the routine metric for monitoring bacterial growth due to its broad applicability in fundamental microbiological research: most bacteria exhibit strong light scattering around the 600 nm wavelength, while commonly used culture media exhibit low background absorption at this wavelength¹⁻⁴. Thus, OD₆₀₀ provides a stable, convenient, and high throughput reference for relative concentration measurements. Crucially, to ensure our inoculum precisely met the CLSI requirement, we did not rely on OD₆₀₀ alone. We performed plate counting to ensure the accuracy of the OD₆₀₀ adjusted bacterial concentration, confirming the methods are reliable and consistent with standard practices. (Page 4, lines 156-157; lines 168-170)

#2. Incubation temp of 37 rather than 35 C is high and may not detect methicillin resistance per CLSI M100.

The authors’ answer: Thank you for your careful attention to detail in reviewing our manuscript. We sincerely apologize for the typographical error regarding the incubation temperature, we mistakenly wrote 37°C instead of the correct 35°C. This error resulted from a clerical oversight. The correct incubation temperature used in our study was indeed 35°C, in accordance with CLSI M100 guidelines for detecting methicillin resistance. This correction does not affect our conclusions, as 35°C is the standardized temperature for antimicrobial susceptibility testing. We have corrected this in the revised manuscript to avoid any confusion. (Page 4, lines 151; lines 174; Page 5, lines 182; lines 193; lines 206; Page 6, lines 223; Page 8, lines 331)

#3. Please provide other data regarding QC (or correlation between phenotypic and molecular AST results) and a more appropriate citation to justify the parameters that were used.

The authors’ answer: Thank you for your comment. To further validate the reliability of our phenotypic antimicrobial susceptibility testing (AST) results and ensure strict quality control (QC), we performed whole-genome sequencing (WGS) on *S. aureus* isolates as required. Our analysis revealed a high consistency between the phenotypic and molecular data. All (100%)

methicillin-resistant strains carried the *mecA* gene. This strong correlation further demonstrates the accuracy of our initially obtained phenotypic test results. We have added a consistency analysis table to the revised manuscript and uploaded the WGS data of 20 MRSA strains and 20 MSSA strains to the Zenodo database (<https://doi.org/10.5281/zenodo.18062160>) as additional quality control evidence. (Page 4, lines 145-147; Table S10)

References:

1. Cui H, Wu Z, Shi X, et al. CS/PVP Hydrogel-Based Nanocapillary for Monitoring Bacterial Growth and Rapid Antibiotic Susceptibility Testing. *ACS SENSORS*. 2024 2024-07-26;9(7):3540-8.
2. Vila-Farres X, Sauve K, Oh J, et al. Rapid bacteriolysis of *Staphylococcus aureus* by lysin exebacase. *MICROBIOL SPECTR*. 2023 2023-10-17;11(5).
3. Zhang Y, Zhang T, Xiao X, et al. CRISPRi screen identifies FprB as a synergistic target for gallium therapy in *Pseudomonas aeruginosa*. *NAT COMMUN*. 2025 2025-07-01;16(1):5870.
4. Jia J, Zheng M, Zhang C, et al. Killing of *Staphylococcus aureus* persisters by a multitarget natural product chrysothymycin A. *SCI ADV*. 2023 2023-08-04;9(31):g5995.

Re: Spectrum01508-25R2 (**Rapid Identification and Methicillin Resistance Test to *Staphylococcus aureus* in Cerebrospinal by MALDI-TOF MS**)

Dear Mr. Mengyu Zhang:

Your manuscript has been accepted, and I am forwarding it to the ASM production staff for publication. Your paper will first be checked to make sure all elements meet the technical requirements. ASM staff will contact you if anything needs to be revised before copyediting and production can begin. Otherwise, you will be notified when your proofs are ready to be viewed.

Sincerely,
Rebecca Yee
Editor
Microbiology Spectrum